# Brain Age Prediction Using 2D Projections Based on Higher-Order Statistical Moments and Eigenslices from 3D Magnetic Resonance Imaging Volumes

**DOI:** 10.3390/jimaging9120271

**Published:** 2023-12-06

**Authors:** Johan Jönemo, Anders Eklund

**Affiliations:** 1Division of Medical Informatics, Department of Biomedical Engineering, Linköping University, 581 83 Linköping, Sweden; 2Center for Medical Image Science and Visualization (CMIV), Linköping University, 581 83 Linköping, Sweden; 3Division of Statistics and Machine Learning, Department of Computer and Information Science, Linköping University, 581 83 Linköping, Sweden

**Keywords:** brain age, 3D CNN, 2D projections, deep learning, principal component analysis, skewness, kurtosis

## Abstract

Brain age prediction from 3D MRI volumes using deep learning has recently become a popular research topic, as brain age has been shown to be an important biomarker. Training deep networks can be very computationally demanding for large datasets like the U.K. Biobank (currently 29,035 subjects). In our previous work, it was demonstrated that using a few 2D projections (mean and standard deviation along three axes) instead of each full 3D volume leads to much faster training at the cost of a reduction in prediction accuracy. Here, we investigated if another set of 2D projections, based on higher-order statistical central moments and eigenslices, leads to a higher accuracy. Our results show that higher-order moments do not lead to a higher accuracy, but that eigenslices provide a small improvement. We also show that an ensemble of such models provides further improvement.

## 1. Introduction

With the availability of large amounts of openly available magnetic resonance imaging (MRI) data and the relative ease of constructing machine learning models, many turn to training such models to estimate various metrics from MRI volumes [1]. One such metric that seems to have physiological significance in a range of conditions is brain age—that is to say, the apparent age estimated from neuroimaging data [2,3,4]. This was presented as an important biomarker in categorizing aging subjects by Cole in 2017 [5] and has since been investigated as a biomarker for different forms of dementia [6], where it seems particularly promising for Alzheimers (the difference in brain age and chronological age was well correlated to severity as measured by tau-protein-binding tracer positron emission tomography (tau-PET) within groups with minor cognitive impairment (MCI) and Alzheimer’s disease (AD)) [7]. Other researchers have suggested that brain age is correlated with hypertension [8] and severity of depression [9,10], and that it is also predictive of the success of certain interventions for chronic pain [11]. Furthermore, an inflated brain age associated with schizophrenia has been shown to be partly reversed at the onset of medication [12,13]. There are a few recent review articles that give a more thorough explanation of the subject [14,15,16].

### 1.1. Related Work on Deep-Learning-Based Brain Age Prediction

There have indeed been many deep learning models for brain age prediction suggested in the recent literature; see Tanveer et al. for a recent review [4]. The goal has often been to minimize the mean absolute error (MAE) between predicted brain age and biological age. More traditional machine learning methods (for example, using the size of different brain regions in a standard regression model) have also been used for predicting brain age [17]. Many of the deep models use 3D convolutional neural networks (CNNs) on whole or possibly down-sampled brain MRI volumes [3,7,18,19,20,21,22], and a large portion of these studies have trained their models with U.K. Biobank data. Such 3D models can be very resource-demanding with respect to processing time and memory consumption, while also suffering from a less mature framework of machine learning software specializing on 3D CNNs and 3D image grid data in general. Other researchers in this field have therefore used slices in one plane from brain volumes in 2D CNNs, weighting the estimates together using some means for the total brain age [23,24,25]. These techniques still use the same amount of data and so can be quite slow, although likely faster than a corresponding 3D CNN. Furthermore, they have an additional problem, which is how to weight all the slices, which in turn also can be performed with a machine learning model or some other algorithm. Also, these models cannot react to patterns occurring perpendicular to the slices.

### 1.2. Our Previous Work

In our previous work, we examined the possibility of assessing brain age using deep learning using a limited amount of two-dimensional images derived from brain volume [26], inspired by Langner et al. [27], instead of using each full 3D volume. The result was a substantially faster training, about 25 min compared to the typical 48 h or more for using a 3D network. Howlever, the accuracy was not as good as that of some of the CNNs that we referred to in our previous paper—the best of them had an MAE of 2.14 [20] compared to about 3.40 with our projection approach—but these methods are hard to compare. For example, the model did not differ only in training and test sets (which, of course, is quite natural): it also differed in that it trained a 3D-CNN for 130 h and used an ensemble of 20 such nets.

The specific images used in our previous work [26] were maps of the mean or standard deviation of values along three axes of the brain volume (transversal, sagittal, coronal). We selected these three projections as they are natural and easy to work with. Furthermore, we believe that, for example, using only one of these projections would remove too much information. The exact nature of these values could conceivably be chosen in any number of ways, but among the ones we have tried, we have found grey matter likelihood as computed using the FSL from T1-structural volumes gives the best results. This is also a very common approach used in studies about predicting brain age (e.g., [3,18,20,28]).

### 1.3. This Work

In this work, we looked at more sources for similar 2D projections that could even better extract the essential information from brain volumes (to further improve the accuracy without increasing training time too much). It should be noted that we here used a looser definition of projection than both its sense in tomography, which corresponds specifically to what we here call the mean channel, and its mathematical meaning of idempotent linear transformation. By projection, we here mean a way to obtain a 2D image from a 3D volume. Figure 1 shows an overview of our 2D projection approach, which, compared to our previous method [26], uses more channels per axis. Specifically, we tried adding skewness and kurtosis to the previous mean and standard deviation maps, thus using up to four 2D projections per axis. Another idea we here pursued was to find essential information in a plane, not by in some way aggregating values along a perpendicular axis but rather by seeing each slice as an example of a two-dimensional representation blueparallel to that particular plane of its volume. In that case, the most informative projections in this set should be available for us to extract by means of principle component analysis (PCA). By seeing each slice (e.g., 256×256 pixels) in a volume as a long vector (e.g., length 65,536), it is possible to use PCA to obtain eigenvectors that capture as much variance as possible (of all slices in the volume). These long eigenvectors can then be reshaped back to what we here call eigenslices. We investigated the representations by up to 16 such eigenslices per axis (perpendicular to the decomposed slices).

## 2. Materials and Methods

### 2.1. Data

Our dataset consists of 29,035 T1-weighted brain volumes from U.K. Biobank [29,30,31,32], which was also used in our previous work [26]. All subjects were scanned using one of four Siemens Skyra 3T scanners with a Siemens 32-channel RF receive head coil, available in Newcastle upon Tyne, Stockport, Reading and Bristol. The sequence used is a 3D MPRAGE sagittal sequence with TI/TR = 880/2000 ms. U.K. Biobank preprocessing of each subject included gradient distortion correction and skullstripping [31]. Due to the skullstripping, all voxels outside the brain were set to zero, meaning that any background noise was ignored. The bias field was already reduced via the on-scanner “pre-scan normalise” option. All volumes were in native space and were not registered to any template as convolutional neural networks do not require objects to be aligned in the way that statistical approaches typically do (and spatial variation can improve generalization). Because of the computation of the likelihood that any particular point inside a voxel is grey matter, all values were clamped to the closed interval from 0 to 1. No further intensity normalization was carried out as preprocessing.

The subjects were divided 70%/15%/15% for training, validation, and testing, respectively. The combined set of training and validation was partitioned in 3 different ways with mutually disjoint validation sets for cross-validation purposes. FSL FAST [33] was used for each skullstripped volume to obtain maps of grey matter. See Figure 2 for one example subject. These grey matter volumes were zero-padded, symmetrically, to match the largest grid size, resulting in volumes of 256 × 256 × 208 voxels with a size of 1 × 1 × 1 mm${}^{3}$.

### 2.2. Higher-Order Statistical Moments

According to our previous investigations, the standard deviation contained more information than the mean, measured by how much the mean average error differed between otherwise identical models with and without one channel [26]. For this reason, it seemed promising to include higher-order statistical moments. We used at most four moment channels. All “intensity” values used from the volumes represent grey matter likelihood. The first two channels were the mean and standard deviation of the voxels lying along a line perpendicular to the projection plane, i.e., the same as in our previous work. For measures of the third and fourth moments (skewness and kurtosis), we calculated the standardized central moments for the voxels lying within the brain, defined as the interval from the first nonzero “intensity” value up to and including the last such value along the aforementioned perpendicular line. Alternatively, for all pairs of coordinates (α,β) in an axis-aligned slice, we only considered the intensities $I_{\alpha \beta }\left(\gamma \right)$ for $\gamma \in \mathrm{hull}(\mathrm{supp}\left(I_{\alpha \beta }\right))$. When few enough values were considered, these higher moments became numerically unstable or even undefined, so we used a value of zero when the path in the brain was sufficiently short (less than 8 voxels). Using $\vec{x}$ for a vector extracted from the brain volume along the dimension that is being reduced, the value in each pixel was computed as:(1)$${\tilde{\mu }}_{k}\left(\vec{x}\right)=n^{k/2-1}\frac{\sum_{i=\alpha }^{\omega }{\left(x_{i}-\mu_{1}\right)}^{k}}{{\left(\sum_{i=\alpha }^{\omega }{\left(x_{i}-\mu_{1}\right)}^{2}\right)}^{k/2}}-C$$
where *k* is 3 or 4, and the following definitions and conditions are in effect:$$\begin{array}{c} \alpha \;\mathrm{is}\;\mathrm{the}\;\mathrm{first}\;\mathrm{position}\;\mathrm{for}\;\mathrm{which}\;x_{i}\neq 0\\ \omega \;\mathrm{is}\;\mathrm{the}\;\mathrm{last}\;\mathrm{position}\;\mathrm{for}\;\mathrm{which}\;x_{i}\neq 0\\ C=\left\{\begin{array}{cc} 0 & k=3\\ 3 & k=4 \end{array}\right.\\ n≜\omega -\alpha +1\\ n\geq 8\;\left(∴\vec{x}\neq \vec{0}\right) \end{array}$$

The constant *C* in (Equation 1), in all likelihood, makes little difference to the CNN since it uses several batch normalization layers. It is, however, common for software libraries to include the 3 for kurtosis, thereby making it the so called excess kurtosis, i.e., kurtosis in excess of that of a normal distribution. An example of the different moments is shown in Figure 3.

### 2.3. Eigenslices

The other sort of image we here employed was produced from each brain volume, which was regarded as a stack of slices in the plane of two coordinate axes, using PCA. The motivation for this is that of all linear bases of dimension *k*, by construction, the one obtained by the first *k* eigenvectors is the one that preserves the most variation in the projected data. It could be noted that this type of 2D image does not have a very intuitive anatomic interpretation. Rather, it represents a way to reduce the amount of data while retaining a large amount of information. The problem of interpretation is thereby deferred to the deep learning network. The assumption is thus that the deep learning model can be trained to use this information, even if a human cannot.

The procedure of generating these slices is performed independently for each subject and each projection axis. If we regard each 2D-slice as a vector (rows or columns could be concatenated), we can assemble these as column vectors into a matrix M. In terms of M, we want to find the eigenvectors with the highest eigenvalues of ${\mathrm{MM}}^{T}$. This is a very large matrix (65,536×65,536 or 53,248×53,248, for projections of 256×256 and 256×208 pixels, respectively, but with low rank (the number of nonzero 2D slices)). We therefore employed the technique used by Sirovich et al. in 1987 and Turk et al. in 1991 in the context of facial recognition [34,35], whereby we use the following relationship:$$\begin{array}{c} M^{T}M{\vec{v}}_{i}=\lambda_{i}{\vec{v}}_{i}\\ ⇓\\ {\mathrm{MM}}^{T}M{\vec{v}}_{i}=M\lambda_{i}{\vec{v}}_{i}\\ ⇕\\ {\mathrm{MM}}^{T}\left(M{\vec{v}}_{i}\right)=\lambda_{i}\left(M{\vec{v}}_{i}\right) \end{array}$$

In other words, if we compute the eigenpairs $\left(\lambda_{i},{\vec{v}}_{i}\right)$ of the much smaller $M^{T}M$ such that $\lambda_{1}\geq \lambda_{2}\geq \ldots \geq \lambda_{n}$, the corresponding $M{\vec{v}}_{i}$ is the eigenvectors originally sought, here called eigenslices. See Figure 4 for eigenslices 1 to 4 for one subject.

### 2.4. Two-Dimensional Projection CNN

Figure 1 shows an overview of our 2D projection approach, where the statistical moments or eigenslices are used for brain age prediction in a 2D CNN with three stacks (one per axis: transversal, sagittal, and coronal). Both the generation of the two-dimensional images and all machine learning models were implemented in and run with Julia version 1.8.5 [36]. In the machine learning parts, we made use of the Julia module Flux version 0.13.11 [37]. The training was performed on an Nvidia RTX 8000 graphics card with 48 GB of memory.

Once the two-dimensional images are obtained, they are cached to permanent storage to obviate the need to compute them repeatedly and fed to a machine learning model with three parallel 2D CNN stacks for the different “projection” planes. The stacks are made up of units of convolution → activation → convolution → batch-normalization → activation → dropout, each of which doubles the number of features and halved the resolution along each axis. It also has a capping module to produce a one-dimensional feature vector, which contains another convolutional layer. The model then aggregates the features from all three views and produces an age estimate. It can use either mean square error or mean absolute error as the loss function for training. The used 2D CNN has 13 convolutional layers with 4 filters in the first layer. For more details, see our prior work [26].

The hyperparameters were optimized manually. Different positions for the dropout layers and different dropout rates comparisons are shown in our earlier article [26]. Optimization was performed using the Adam optimiser with a learning rate of 0.003 [38] and a batch size of 32. All models were trained for 400 epochs, but the model state after training was chosen to be the one with the best validation accuracy, as in early stopping. The constant epoch training was performed mainly for more complete speed metrics. Seeing as these models take relatively little time to train and that we already had several of them trained and saved, we also looked at if the models could be used in ensemble to further improve the accuracy. The used code is available at (accessed on 1 December 2023) https://github.com/emojjon/eigen-moments-brain-age.

## 3. Results

The network was trained repeatedly with different combinations of in-channels. Every variation was furthermore trained several times in order to estimate a measure of dispersion (except for the ensembles, as this would require much more time). All trainings here used the corresponding channels for all three projections (although having different combinations of channels per projection is also supported). Channels 1 to *n* were used, or just channel *n* for *n* ∈ {1, …, 8}, for the eigenslices and *n*∈ {1, …, 4} for the moments.

It should be said that the trainings were initially run—and possibly rerun—to give an overview to us as researchers and possibly to suggest improvements (for convenience, the models remained mathematically equivalent). After that, new trainings were run so that at least four trainings exists for every combination of parameters here presented. In the cases where more trainings had already been run, all were kept, as having more measurements does not per se affect the expected value of the mean or the standard deviation, if correctly computed.

The results are visualized in Figure 5 and presented in a more comprehensive form in Table 1. Using higher-order moments does not seem to improve the accuracy compared to using mean and standard deviation. As expected, when only using a single eigenslice, the accuracy deteriorates for higher-order eigenslices, as these eigenslices represent less and less of the variance. Using the first two eigenslices leads to a slightly better accuracy (MAE of 3.36 years) compared to using mean and standard deviation (MAE of 3.47 years). Somewhat surprisingly, the performance is reduced when using increasingly more eigenslices together.

Some further variations were preliminarily evaluated but discontinued because they did not provide any advantages. This included all runs involving eigenslices 9 to 16 and—perhaps surprisingly—runs with mean absolute error as the loss function.

The training times for these models were comparatively short. Both the early stopping time and the time to train for 400 epochs, which was considered to be enough to train any of the models, are listed in Table 1. A leap in training time was typically seen between using three channels per projection axis and using four. This is due to the fact that the model estimates the amount of GPU memory needed to fit the training data and resorts to a strategy of uploading smaller parts to GPU memory during each epoch of training should this amount not be available.

We also tried to use four trained models in ensembles for all models trained with the 2–5 first eigenchannels. Only one ensemble of each kind was evaluated, where there is no standard deviation for the MAE. The corresponding dispersion measure for the constituent models transpired from the table. Clearly, the accuracy improved compared to using a single model. For example, the MAE decreased from 3.36 to 3.18 years when using the two first eigenslices.

## 4. Discussion

This is the continuation of our previous work [26], where we investigated a similar approach but using only the mean and standard deviation over each dimension to obtain six channels. In this work, we included more channels to feed into the network to improve accuracy without increasing training time too much. We added skewness and kurtosis to the projections with mean and standard deviation. We also investigated using eigenslices from the PCA of one “stack” of slices per dimension and subject.

The measure we studied here was brain age, and we trained our models with the assumption that all used subjects should be healthy and thus present a brain age equal to their biological age. It should therefore be noted that less than perfect correlation between brain age and biological age in healthy subjects, as well as a less than perfect classification of who is “healthy”, would be part of the error of the models. Furthermore, even a model that could predict the biological age perfectly would have an MAE of 0.25 years because of the rounding of the recorded ages to whole years (for anonymization purposes). All of this taken together means that as we refine our models, the residual deviation from a perfect result should be compared to that of an unknown “best possible result” rather than zero.

For what we here chose to call the “moment” channels (because they largely represent the first through fourth central moments of the grey matter likelihood along an axis-aligned path), the results are hard to interpret. Mainly we saw that the skewness channel seems to perform worse than the others, not only alone: it also seems to confuse models trained with channels 1 to 3 (though strangely not channels 1 to 4).

With the eigenslices, we noticed how each consecutive slice by itself (applied in all three directions) leads to a worse prediction of brain age than the one before it. This is expected as the eigenslices are sorted by their eigenvalues, which in turn should give a measure of the explanatory power of that slice. In the case of slices 1 to *n*, i.e., all slices up to number *n*, we noticed a much more even curve although there did not seem to be any benefit to including more than two or perhaps three slices in each dimension.

In general, the explanatory power of the eigenslices tapers off quite fast. This is probably because, although (roughly) corresponding eigenslices might be generated for different brains, as we proceed down the stack, the eigenvalues lie closer and closer to each other and hence the order of corresponding eigenslices can change. This would make the data much harder for our model to learn from. A possible development to the technique could be to change the order of some eigenslices so as to increase their correspondence. The exact algorithm would have to be investigated further. Another solution can in theory be to perform PCA on all subjects concurrently. However, this may be very computationally expensive and would require an extra step to obtain subject specific eigenslices from the group eigenslices. At this point, we have not verified that this extra step is possible, but it would be inspired by the dual regression (spatial and temporal) approach used for 4D functional MRI data [39]. Another idea is to perform PCA on all 20,325 training volumes to first obtain eigenvolumes and then project the volume from each subject on the *k* first eigenvolumes. Then, one could use these coordinates directly or use them to construct a new volume and calculate 2D projections from it.

We also looked at making ensembles of trained models. The advantage of this is contingent on how highly the errors in different models are correlated. Making ensembles of four models—each with eigenslice channels 1 to *n*—for a range of *n*s, we could see a clear improvement in the accuracy.

The measured times are potentially not representative for a model working under optimal conditions because no provisions have been made for picking and mixing channels, only for limiting them to the *n* first ones along each axis. This means that even if only one channel is used for the training, the lower-numbered channels would still be loaded into memory and either past in its entirety or shuttled on demand in little pieces to the GPU. Should one need to do this on a regular basis, one could write a short definition of a projection containing a minimal set of channels to a Julia file and include it like any other projection, specifying a cache name and making sure to provide the bundle of channels in the corresponding directory or include instructions for how to generate them and let them be created on demand.

## 5. Conclusions

To summarize, to use higher-order moments does not improve the results obtained in our previous study [26], where only mean and standard deviation were used. To instead use the first two eigenslices provides a small improvement, from an MAE of 3.47 to 3.36 years, compared to using mean and standard deviation (but we did not test for statistical significance). It is possible that somehow sorting the eigenslices or using eigenvolumes can further improve the results. Using an ensemble of models provides further improvement, from an MAE of 3.36 to 3.18 years, while the total training time is still much shorter compared to that of 3D CNNs.

## Figures and Tables

**Figure 1 jimaging-09-00271-f001:**
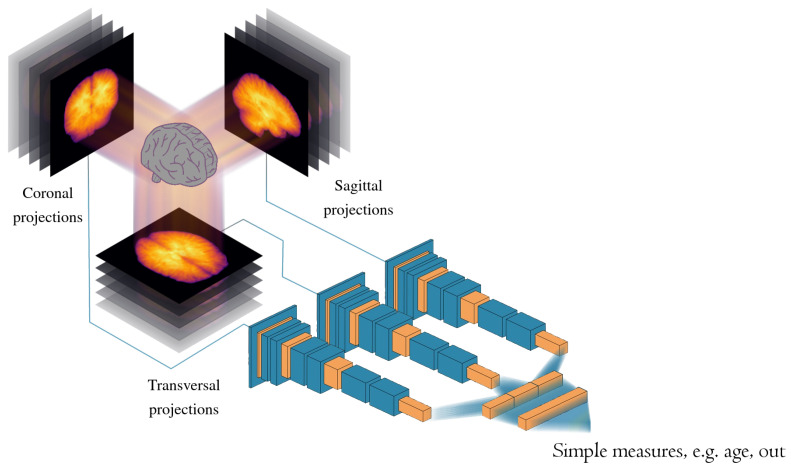
A conceptual illustration of our machine learning model used for brain age prediction. From each brain volume, a number of 2D images (or projections) are created by “collapsing” each of the three spatial dimensions. Two methods of collapsing each dimension were investigated in this work: calculating different statistical moments along each axis and calculating so-called eigenslices perpendicular to each axis. The images are passed to one of three stacks of convolutional and auxiliary layers corresponding to what dimension is missing. The extracted features from the three stacks are concatenated and input to a small dense network, which produces the final brain age estimate.

**Figure 2 jimaging-09-00271-f002:**
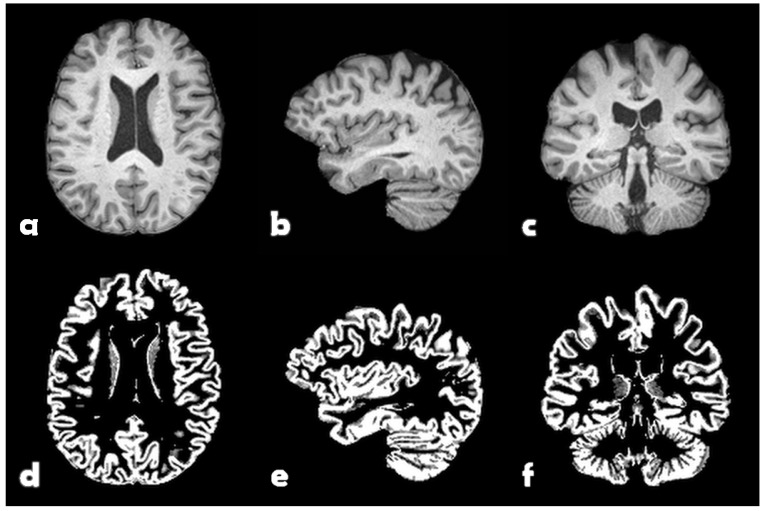
One example subject from U.K. Biobank. Top: Preprocessed T1-weighted volume: (**a**) transversal slice, (**b**) sagittal slice, and (**c**) coronal slice. Bottom: Grey matter probability map: (**d**) transversal slice, (**e**) sagittal slice, and (**f**) coronal slice.

**Figure 3 jimaging-09-00271-f003:**
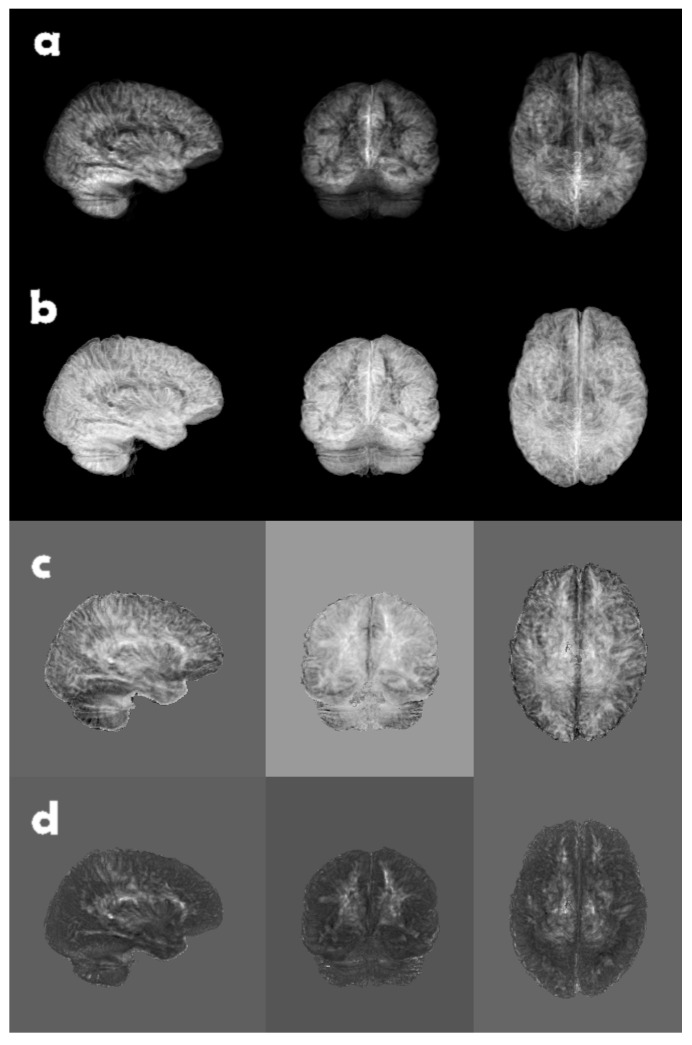
A brain volume of grey matter likelihood, reduced along each coordinate axis, using top (**a**) mean, (**b**) standard deviation, (**c**) skewness, and (**d**) excess kurtosis. Due to the fact that skewness and excess kurtosis can be both positive and negative in conjunction with the normalization of the grey scale, the backgrounds appear as different shades of grey.

**Figure 4 jimaging-09-00271-f004:**
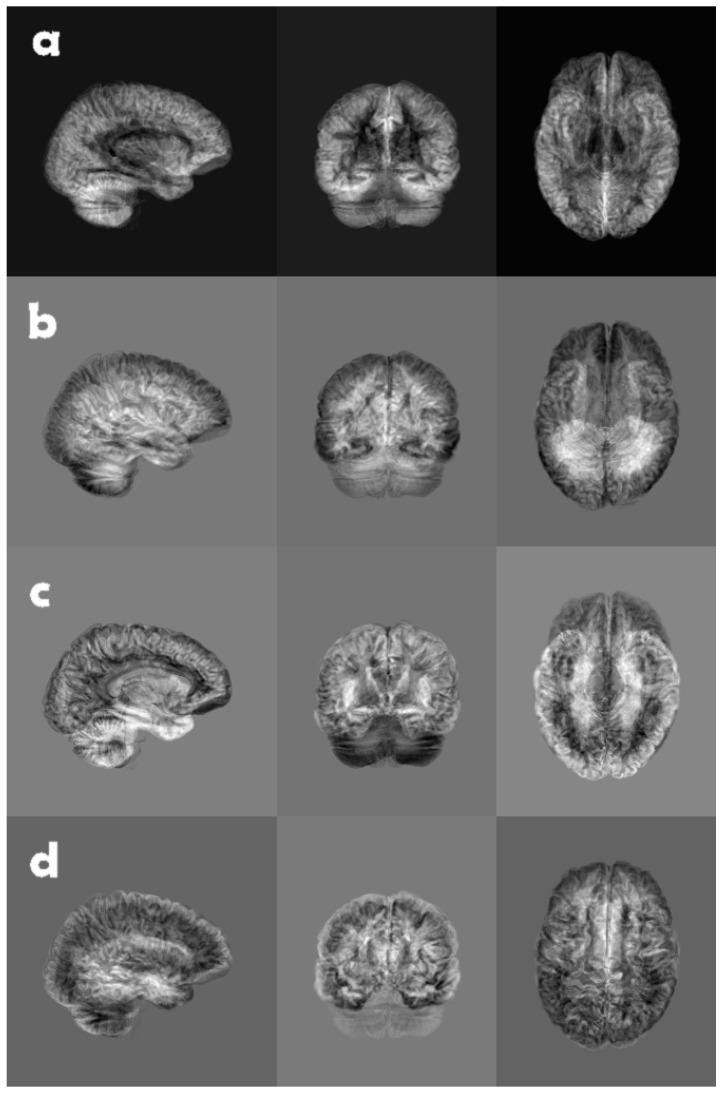
A brain volume of grey matter likelihood, reduced along each coordinate axis, using, from top to bottom, (**a**) eigenslice 1, (**b**) eigenslice 2, (**c**) eigenslice 3, and (**d**) eigenslice 4. The eigenslices were obtained using principal component analysis, where each slice in a volume was seen as a long vector. The eigenslices were calculated for one volume (subject) at a time.

**Figure 5 jimaging-09-00271-f005:**
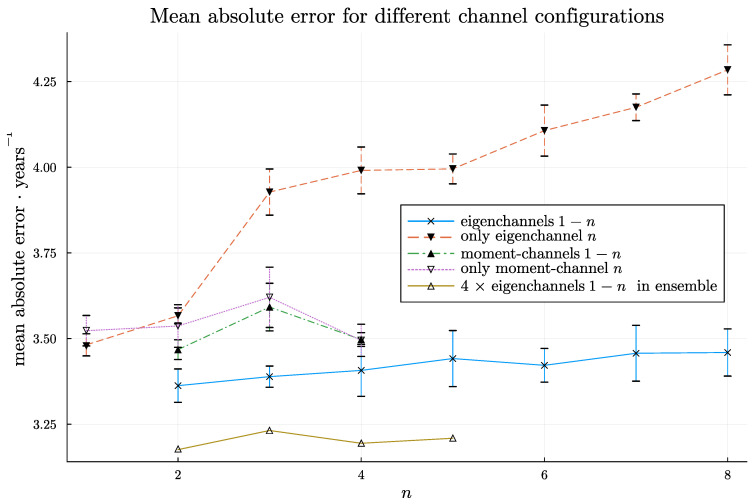
The average MAE for models trained with *n* channels (along each projection axis) or only the channel numbered *n* (along each projection axis). The standard deviation was not calculated for the ensembles, but it should be expected to be between $\frac{1}{\sqrt{4}}$ and 1 times that of the eigenchannels 1 to *n* series for purely algebraic reasons. These values are also shown in Table 1.

**Table 1 jimaging-09-00271-t001:** Results for our 2D projection approach regarding number of training subjects (N), brain age test accuracy (mean absolute error (MAE) in years, RMSE in parenthesis), and training time.The means and standard deviations in the eigenchannel parts were derived from sets of at least 4 trainings (if more than 4 such trainings, for various reasons, had been run, these values were also included to obtain better estimates); no set contained more than 11 trainings), whereas the corresponding values for the moment parts were derived from 4 trainings per row. Even though several publications use the U.K. Biobank data, a direct comparison of the test accuracy was not possible as different test sets, in terms of size and the specific subjects, were used. The training times refer to running on a single GPU. The training times are presented for early stopping and for the full 400 epochs in parentheses.

Input	No. Subjects	Test Accuracy	Parameters	Training Time
**”moment” channels**				
**from 1 to n**				
2	20,325	3.47 ±0.029 (4.38 ±0.049)	2,009,369	28 m 44 s (3 h 19 m)
3	20,325	3.59 ±0.069 (4.51 ±0.079)	2,009,477	1 h 27 m (5 h 7 m)
4	20,325	3.50 ±0.020 (4.44 ±0.031)	2,009,585	1 h 50 m (21 h 23 m)
**”moment” channel**				
**n only**				
1	20,325	3.52 ±0.044 (4.45 ±0.042)	2,009,261	20 m 51 s (2 h 49 m)
2	20,325	3.54 ±0.062 (4.46 ±0.071)	2,009,261	17 m 9 s (3 h 3 m)
3	20,325	3.62 ±0.088 (4.58 ±0.081)	2,009,261	2 h 28 m(3 h 48 m)
4	20,325	3.49 ±0.047 (4.44 ±0.040)	2,009,261	1 h 39 m (20 h 44 m)
**“eigenchannels”**				
**from 1 to n**				
2	20,325	3.36 ±0.049 (4.25 ±0.055)	2,009,369	19 m 53 s (3 h 9 m)
3	20,325	3.39 ±0.031 (4.28 ±0.040)	2,009,477	28 m 5 s (4 h 19 m)
4	20,325	3.41 ±0.076 (4.32 ±0.088)	2,009,585	1 h 59 m (17 h 5 m)
5	20,325	3.44 ±0.082 (4.35 ±0.096)	2,009,693	3 h 57 m (1 d 7 h)
6	20,325	3.42 ±0.049 (4.34 ±0.071)	2,009,801	3 h 23 m (1 d 13 h)
7	20,325	3.46 ±0.081 (4.37 ±0.093)	2,009,909	6 h 1 m (2 d 2 h)
8	20,325	3.46 ±0.069 (4.38 ±0.092)	2,010,017	6 h 8 m (2 d 1 h)
**“eigenchannel”**				
**n only**				
1	20,325	3.48 ±0.032 (4.39 ±0.044)	2,009,261	21 m 8 s (2 h 52 m)
2	20,325	3.57 ±0.023 (4.50 ±0.025)	2,009,261	14 m 20 s (2 h 57 m)
3	20,325	3.93 ±0.067 (4.94 ±0.072)	2,009,261	20 m 34 s (4 h 39 m)
4	20,325	3.99 ±0.068 (5.03 ±0.082)	2,009,261	1 h 52 m (19 h 35 m)
5	20,325	4.00 ±0.044 (5.02 ±0.056)	2,009,261	1 h 49 m (1 d)
6	20,325	4.11 ±0.074 (5.18 ±0.097)	2,009,261	2 h 22 m (1 d 7 h)
7	20,325	4.17 ±0.039 (5.22 ±0.024)	2,009,261	4 h 39 m (2 d 20 h)
8	20,325	4.28 ±0.073 (5.36 ±0.090)	2,009,261	11 h 57 m (2 d 11 h)
**“eigenchannels”**				
**1 to n**				
**ensembles of 4**				
2	20,325	3.18 (4.02)	8,037,476	N/A
3	20,325	3.23 (4.09)	8,037,908	N/A
4	20,325	3.19 (4.06)	8,038,340	N/A
5	20,325	3.21 (4.07)	8,038,772	N/A

## Data Availability

The data used in this work are available through U.K. Biobank, accessed on 1 September 2021, https://www.ukbiobank.ac.uk/.

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
