# Peer review of "Brain Age Prediction Using 2D Projections Based on Higher-Order Statistical Moments and Eigenslices from 3D Magnetic Resonance Imaging Volumes"

_2313-433X, 2023, doi:10.3390/jimaging9120271_

Round 1
Reviewer 1 Report
Comments and Suggestions for Authors
This research explores the use of deep learning to predict brain age from 3D MRI scans. It investigates the impact of utilizing 2D projections, specifically higher-order moments and eigenslices, as opposed to full 3D volumes to expedite training. The study concludes that while higher-order moments do not significantly improve accuracy, eigenslices offer a modest enhancement. Additionally, combining an ensemble of these models leads to further improvement.
This work serves as a continuation of a previously published manuscript on the same topic. The earlier study explored the use of first and second moments (mean and standard deviation) to generate 2D projections from 3D volumes. However, the innovations presented in the current work are somewhat limited, given that only additional projections have been introduced in comparison to the previous research, and the results demonstrate a high degree of similarity. Furthermore, the manuscript suffers from various issues and limitations and necessitates substantial revisions.
Detailed Points:
1. The Introduction section is missing a clear declaration of the research objectives, making it challenging to follow the text and comprehend the authors' intent. The authors should explicitly outline how the current work differs from the previous one.
2. The organization of the Methods section requires improvement. For instance, Section 2.2 is titled "Deep learning," but there is a lack of description about deep learning methods. Additional details regarding CNN architecture, training, and other information (such as optimizer, number of epochs, hyperparameters, and image normalization) should be included in section 2.5. The inclusion of an illustrative figure could enhance the section's clarity.
3. Traditional image pre-processing for brain age estimation from T1w-MRI data does not appear to have been applied. Previous research has indicated that the preprocessing stage for neuroimaging data is crucial for developing a brain age estimation framework [ref 4]. Have the authors considered implementing the VBM framework, which includes bias field correction, intensity, and spatial normalization, before computing 2D projections?
4. The method for calculating the standard deviations reported in Table 1 needs further clarification. The text mentions that "standard deviations in the eigenchannel parts are derived from 4 - 11 trainings per row, whereas the corresponding values for the moment parts are derived from 4 trainings per row." It is essential to explain the source of these different numbers of trainings for the two cases and why repeated trainings were not conducted to assess standard deviation in ensemble learning.
Minor Issues:
5. Figure citations in the text should be checked for correct numbering.
6. The sentence at page 1, lines 23-27 should be rephrased for clarity.
7. The last sentence in the Introduction that discusses common approaches used in predicting brain age requires a reference.
8. Figures 1 and 2 should include numerical or caption labeling (e.g., a, b, c, or mean, standard deviation, skewness, kurtosis) to enhance readability.
Comments on the Quality of English LanguageEnglish is fine, only minor editing is required.
Author Response
This research explores the use of deep learning to predict brain age from 3D MRI scans. It investigates the impact of utilizing 2D projections, specifically higher-order moments and eigenslices, as opposed to full 3D volumes to expedite training. The study concludes that while higher-order moments do not significantly improve accuracy, eigenslices offer a modest enhancement. Additionally, combining an ensemble of these models leads to further improvement.
This work serves as a continuation of a previously published manuscript on the same topic. The earlier study explored the use of first and second moments (mean and standard deviation) to generate 2D projections from 3D volumes. However, the innovations presented in the current work are somewhat limited, given that only additional projections have been introduced in comparison to the previous research, and the results demonstrate a high degree of similarity. Furthermore, the manuscript suffers from various issues and limitations and necessitates substantial revisions.
Detailed Points:
1. The Introduction section is missing a clear declaration of the research objectives, making it challenging to follow the text and comprehend the authors' intent. The authors should explicitly outline how the current work differs from the previous one.
We have added an additional paragraph in the introduction, which summarizes the research objectives and explains how this work differs compared to our previous work. It also explains eigenslices.
2. The organization of the Methods section requires improvement. For instance, Section 2.2 is titled "Deep learning," but there is a lack of description about deep learning methods. Additional details regarding CNN architecture, training, and other information (such as optimizer, number of epochs, hyperparameters, and image normalization) should be included in section 2.5. The inclusion of an illustrative figure could enhance the section's clarity.
We have added information regarding optimizer, number of epochs, batch size, hyperparameters, and image normalization. We have merged "deep learning" and "2D CNN sections".
We have also added an illustrative figure which shows an overview of the used network.
3. Traditional image pre-processing for brain age estimation from T1w-MRI data does not appear to have been applied. Previous research has indicated that the preprocessing stage for neuroimaging data is crucial for developing a brain age estimation framework [ref 4]. Have the authors considered implementing the VBM framework, which includes bias field correction, intensity, and spatial normalization, before computing 2D projections?
The images in UK biobank have already been preprocessed (e.g. skullstrip, gradient distortion correction), we have now added information about this in the Data section. The bias field has already been reduced via the on-scanner “pre-scan normalise” option used when scanning. We used the volumes without registration (spatial normalization), to provide more spatial variation for our network (so it can generalize better).
4. The method for calculating the standard deviations reported in Table 1 needs further clarification. The text mentions that "standard deviations in the eigenchannel parts are derived from 4 - 11 trainings per row, whereas the corresponding values for the moment parts are derived from 4 trainings per row." It is essential to explain the source of these different numbers of trainings for the two cases and why repeated trainings were not conducted to assess standard deviation in ensemble learning.
We have added a new paragraph under results which explains why the eigenchannel models were trained a larger number of times.
To assess standard deviation for ensembles would take much more time, as each ensemble requires a number of trainings.
Minor Issues:
5. Figure citations in the text should be checked for correct numbering.
Yes this is a problem with the Latex package, we have now hardcoded the figure numbers.
6. The sentence at page 1, lines 23-27 should be rephrased for clarity.
We have rewritten this sentence
7. The last sentence in the Introduction that discusses common approaches used in predicting brain age requires a reference.
We have added two references.
8. Figures 1 and 2 should include numerical or caption labeling (e.g., a, b, c, or mean, standard deviation, skewness, kurtosis) to enhance readability.
We have added a, b, c, d to these figures.
Reviewer 2 Report
Comments and Suggestions for Authors
This MS describes an attempt to reduce the time necessary predict brain age from 3D volume magnetic resonance (MR) images using deep learning by first transforming the data into 2D using eigenslices. AI and deep learning (DL) are of great interest to the MRI community today, in part because of their popularity. Unfortunately, it is little understood by the majority of the MRI community, as MRI is to the AI/DL community. Therefore, it is important for an MRI-AI/DL paper to be clearly written to advance AI/DL in the MR community.
The following points should be addressed in this MS to help advance this cause.
1.) There is no mention of the acquisition parameters used in the UK Biobank other then using a T1-weighted image. A T1-weighted image is a very general term for a type of MR image recorded using a repetition time (TR) < T1 and an echo time (TE) << T2. Image contrast is primarily dependent on the spin lattice relaxation time, but not completely. There is some dependence on T2 and the spin density.) These definitions leave a great deal of latitude in the choice of TR and TE and hence variability in the image contrast, especially at the level used by AI/DL. Variability could easily account for any changes in moments.
2.) The process of creating the eigenslices is not clear, especially to an MR scientist. Therefore, an important audience is lost. What exactly is an eigenslice? What does it represent in the way of real world attributes?
3.) All MR images have noise, visible in the background and regions of the anatomy devoid of tissue. How is this dealt with? Was a signal threshold set?
4.) Figure 3. It would be clearer to the reader to see markers with error bars, instead of just error bars. Additionally, the lines used for two of the plots are not discernable on my output.
5.) All the references to figures are missing the figure numbers.
Author Response
This MS describes an attempt to reduce the time necessary predict brain age from 3D volume magnetic resonance (MR) images using deep learning by first transforming the data into 2D using eigenslices. AI and deep learning (DL) are of great interest to the MRI community today, in part because of their popularity. Unfortunately, it is little understood by the majority of the MRI community, as MRI is to the AI/DL community. Therefore, it is important for an MRI-AI/DL paper to be clearly written to advance AI/DL in the MR community.
The following points should be addressed in this MS to help advance this cause.
1.) There is no mention of the acquisition parameters used in the UK Biobank other then using a T1-weighted image. A T1-weighted image is a very general term for a type of MR image recorded using a repetition time (TR) < T1 and an echo time (TE) << T2. Image contrast is primarily dependent on the spin lattice relaxation time, but not completely. There is some dependence on T2 and the spin density.) These definitions leave a great deal of latitude in the choice of TR and TE and hence variability in the image contrast, especially at the level used by AI/DL. Variability could easily account for any changes in moments.
We have added details about the used MR scanner, the coil and the sequence parameters. All subjects have been scanned using the same type of scanner.
When we talk about moments we mean statistical moments, not magnetic moments. We have added "statistical" in the title to prevent confusion.
2.) The process of creating the eigenslices is not clear, especially to an MR scientist. Therefore, an important audience is lost. What exactly is an eigenslice? What does it represent in the way of real world attributes?
The process is described in mathematical terms. We have now added a textual description in the last paragraph of the introduction, and added more text in the section about eigenslices.
3.) All MR images have noise, visible in the background and regions of the anatomy devoid of tissue. How is this dealt with? Was a signal threshold set?
To the best of our knowledge no signal threshold was set in UK Biobank's preprocessing. The noise level is probably so low that it does not affect further processing.
4.) Figure 3. It would be clearer to the reader to see markers with error bars, instead of just error bars. Additionally, the lines used for two of the plots are not discernable on my output.
This has been added.
5.) All the references to figures are missing the figure numbers.
Yes this is due to an issue with the latex template, we have now hardcoded the figure numbers.
Reviewer 3 Report
Comments and Suggestions for Authors
Although the abstract promises a very interesting paper, there are some parts of the submitted manuscript that are rather weak.
i) Unfortunately, the authors do not review related and previous work. In the Introduction section, one paragraph deals with the relevant state-of-the-art in several sentences briefly. I think that a scientific publication should deal with the relevant state-of-the-art more carefully than those of this manuscript. I think a separate related work section could solve this problem.
ii) In section 2.1, the authors give an overview of the used benchmark databases. I think several images from these databases should be inserted into the manuscript as an illustration for readers coming from different backgrounds. Further, the authors state that "The brain volumes are reduced to two-dimensional images in the transversal, coronal and sagital planes." A brief explanation of transversal, coronal, and sagital planes should be inserted here. Probably, not everybody knows the exact meaning of these medical expressions.
iii) Several references in the Latex code may be wrong, because several ??s are in the text.
iv) Sections 2.3-2.5 are unfortunately very weak. The structure of the CNN was totally unclear to me. Further, it was also unclear how this CNN was trained. I mean how higher order moments and eigenslices were incorporated into the model or the training data. The authors could illustrate these sections with informative flowcharts or figures.
v) The related work section is also rather weak. First, the authors should define carefully what the evaluation metrics is. Second, it is totally unclear how the state-of-the-art was improved. Did the authors achieve significant results? Could you verify this with significance tests?
Author Response
Although the abstract promises a very interesting paper, there are some parts of the submitted manuscript that are rather weak.
i) Unfortunately, the authors do not review related and previous work. In the Introduction section, one paragraph deals with the relevant state-of-the-art in several sentences briefly. I think that a scientific publication should deal with the relevant state-of-the-art more carefully than those of this manuscript. I think a separate related work section could solve this problem.
We have divided the introduction into different sections, and made the related work paragraph longer, and also mentioned more traditional machine learning.
ii) In section 2.1, the authors give an overview of the used benchmark databases. I think several images from these databases should be inserted into the manuscript as an illustration for readers coming from different backgrounds.
We have included one example subject, showing the brain volume and the calculated gray matter probability.
Further, the authors state that "The brain volumes are reduced to two-dimensional images in the transversal, coronal and sagital planes." A brief explanation of transversal, coronal, and sagital planes should be inserted here. Probably, not everybody knows the exact meaning of these medical expressions.
This can now be seen in the new Figure 1 (showing an overview of our network) and in the new Figure 2 (showing one example subject).
iii) Several references in the Latex code may be wrong, because several ??s are in the text.
Yes this is due to the used Latex package, we have now hard coded the figure numbers.
iv) Sections 2.3-2.5 are unfortunately very weak. The structure of the CNN was totally unclear to me. Further, it was also unclear how this CNN was trained. I mean how higher order moments and eigenslices were incorporated into the model or the training data. The authors could illustrate these sections with informative flowcharts or figures.
The new figure 1 shows an overview of our used approach. We have also added more details about the CNN and the training, as it was also requested by reviewer 1. We have merged "deep learning" and "2D CNN" to one section.
v) The related work section is also rather weak. First, the authors should define carefully what the evaluation metrics is.
We have made the related work section longer, and mentioned that most papers focus on MAE.
Second, it is totally unclear how the state-of-the-art was improved. Did the authors achieve significant results? Could you verify this with significance tests?
The focus of our work is much faster training, while still obtaining decent predictions. Since our improvements were not so large we did not test for statistical significance. We have extended the conclusions section to make it more clear what the improvements are in this paper.
Round 2
Reviewer 1 Report
Comments and Suggestions for Authors
I appreciated the modifications made by the authors. The manuscript is now more clear and the quality is improved. I only suggest to ahve a final check to English language throughout the whole manuscript.
Comments on the Quality of English LanguageEnglish is fine, just a final check is required for minor editing
Author Response
We have went through the manuscript and made some minor language improvements.
Reviewer 2 Report
Comments and Suggestions for Authors
The description of 1.3 This Work and 2.1 Data are a definite improvement over the first version, such that I am now able to ask questions.
Since this method is compared to a previous one, were the same data sets used for both?
What is a projection? Projection has a very specific connotation/meaning in tomographic imaging. Is it a literal projection where say the projection (P) in the x,y plane is the sum of all the z voxel signals (S) at a fixed x,y?
P(x,y) = Sz (S(x,y,z))
Or are they the mean, standard deviation, skewness, and excess kurtosis in the projection direction? How are zero voxel values (voxels outside the brain stripped by the skull stripping process) treated for calculation of mean, standard deviation, skewness, and excess kurtosis channels? Or are the channels the first k eigenvectors from the PCA? What is the physical significance of these four channels? Can other normal variations in the brain result in altered channel values unrelated to age? Why are sagittal, coronal, and axial “projections” used?
Thank you for including a description of the data bank images. Were these images recorded on a single scanner at a single site, or the same type scanner at multiple sites.
In the first review the AUs were asked
3.) All MR images have noise, visible in the background and regions of the anatomy devoid of tissue. How is this dealt with? Was a signal threshold set?
To which they responded.
To the best of our knowledge no signal threshold was set in UK Biobank's preprocessing. The noise level is probably so low that it does not affect further processing.
If the biobank images had skull stripping, everything outside the skull is typically set to zero. It might appear as though there is no noise because of a zeroed background, but this noise is in the image data also. The noise and SNR can be readily calculated from the pre-processed (unstripped) MR images. It seems as though this would be important for validation of the technique. There are digital brain phantoms that could be used to test the procedure and how noise influences the results.
Author Response
The description of 1.3 This Work and 2.1 Data are a definite improvement over the first version, such that I am now able to ask questions.
Since this method is compared to a previous one, were the same data sets used for both?
Yes the same data were used for this paper and our previous paper. We have now clarified this in the paper.
---
What is a projection? Projection has a very specific connotation/meaning in tomographic imaging. Is it a literal projection where say the projection (P) in the x,y plane is the sum of all the z voxel signals (S) at a fixed x,y?
P(x,y) = Sz (S(x,y,z)). Or are they the mean, standard deviation, skewness, and excess kurtosis in the projection direction?
We use projection in a loose definition, meaning any way to obtain a 2D image from a 3D volume (i.e. mean or some other moment in the projection direction, or using eigenslices in the same directions). We have now mentioned this in the paper.
---
How are zero voxel values (voxels outside the brain stripped by the skull stripping process) treated for calculation of mean, standard deviation, skewness, and excess kurtosis channels?
As mentioned in 2.2, for mean and standard deviation the background zero voxels are not a problem. Skewness and kurtosis are more numerically unstable, and were not calculated for voxels with a short path through the brain. This is already mentioned in 2.2
"When few enough values are considered, these higher moments get numerically unstable or even
undefined, wherefore we used the value zero when the path in the brain was sufficiently short (less than 8 voxels)."
---
Or are the channels the first k eigenvectors from the PCA?
Yes our second approach in this paper is to use the first k eigenvectors (eigenslices) as input channels, instead of using statistical moments. This approach seems to work a little better.
---
What is the physical significance of these four channels?
We think it is difficult to think of eigenslices in terms of physical significance. They should rather be seen as images that capture most of the variance across slices in a volume (for one direction at a time). In the case of eigenfaces, which inspired our work, eigenfaces represent the most common face and the most common variations of faces that exist for a certain collection of face images.
---
Can other normal variations in the brain result in altered channel values unrelated to age?
We assume that this is possible, and that the challenge of our deep network is to understand which altered channel values that are really related to a change in age.
---
Why are sagittal, coronal, and axial “projections” used?
We used sagittal, coronal, and axial projections because it is a natural and easy way of obtaining 2D images from a 3D volume. It would be much harder to obtain 2D images through projections along some arbitrary directions. We also reason that for example only using axial projections will remove too much information from each volume. We have now clarified this under 1.2.
---
Thank you for including a description of the data bank images. Were these images recorded on a single scanner at a single site, or the same type scanner at multiple sites.
According to the UK biobank homepage, the same type of scanner is used at the following sites. We have now added this information in 2.1
We have imaging centres in Newcastle upon Tyne, Stockport, Reading and Bristol.
---
In the first review the AUs were asked
3.) All MR images have noise, visible in the background and regions of the anatomy devoid of tissue. How is this dealt with? Was a signal threshold set?
To which they responded.
To the best of our knowledge no signal threshold was set in UK Biobank's preprocessing. The noise level is probably so low that it does not affect further processing.
If the biobank images had skull stripping, everything outside the skull is typically set to zero. It might appear as though there is no noise because of a zeroed background, but this noise is in the image data also. The noise and SNR can be readily calculated from the pre-processed (unstripped) MR images. It seems as though this would be important for validation of the technique. There are digital brain phantoms that could be used to test the procedure and how noise influences the results.
Yes we apologize, we misunderstood the question, due to the skullstrip everything outside the brain is set to 0. We have added this to the paper.
"Due to the skullstripping, all voxels outside the brain have been set to zero, meaning that any background noise is ignored."
Reviewer 3 Report
Comments and Suggestions for Authors
Based on the authors' answer and the improvements carried out by the authors, the manuscript can be accepted now.
Author Response
Thank you
Round 3
Reviewer 2 Report
Comments and Suggestions for Authors
An improvement over the earlier versions. Thank you for addressing my concerns.